# Anterior and posterior ratio of corneal surface areas: A novel index for detecting early stage keratoconus

**Motohiro Itoi[1], Koji Kitazawa[1,2]\*, Isao Yokota[3], Koichi Wakimasu[2], Yuko Cho[1], Yo Nakamura[1], Osamu Hieda[1], Satoshi Teramukai[3], Shigeru Kinoshita[4], Chie Sotozono[1]**

**1** Department of Ophthalmology, Kyoto Prefectural University of Medicine, Kyoto, Japan, **2** Department of Ophthalmology, Baptist Eye Institute, Kyoto, Japan, **3** Department of Biostatistics, Kyoto Prefectural University of Medicine, Kyoto, Japan, **4** Department of Frontier Medical Science and Technology for Ophthalmology, Kyoto Prefectural University of Medicine, Kyoto, Japan

\* kkitazaw@koto.kpu-m.ac.jp

**Data Availability Statement:** All relevant data are within the manuscript and its Supporting Information files.

## Abstract

### Purpose

To evaluate the diagnostic ability of the ratio of anterior and posterior corneal surface areas (As/Ps) comparing with other keratoconus screening indices in distinguishing forme fruste keratoconus (FFKC) from normal eyes.

### Methods

In this comparative study, 13 eyes of 13 patients with FFKC, 29 eyes of 29 patients with keratoconus (KC) and 88 eyes of 88 patients with normal subjects were involved.

The As/Ps measured by the anterior segment optical coherence tomography (AS-OCT) and other indices measured by AS-OCT and rotating Scheimpflug–based corneal tomography were evaluated. The area under receiver-operating-characteristics (AU-ROC) was calculated to assess the diagnostic ability in discriminating FFKC from normal eyes.

### Results

The As/Ps, the Belin/Ambrosio display enhanced ectasia total derivation value (BAD-D) and posterior and anterior elevation values showed the AU-ROC 0.9 or more in differentiating FFKC from normal eyes (0.980, 0.951, 0.924 and 0.903, respectively). The sensitivity and specificity were 0.92 and 0.96 for the As/Ps, 1.00 and 0.90 for BAD-D, 0.85 and 0.86 for posterior elevation value, and 0.85 and 0.96 for anterior elevation value, respectively.

### Conclusions

Among the several indices for keratoconus screening which we evaluated, the As/Ps obtained by AS-OCT had the large AU-ROC with high sensitivity and specificity in detecting FFKC, which was comparable with BAD-D obtained by rotating Scheimpflug–based corneal

**Funding:** Koji Kitazawa received Grant-in-Aid for Young Scientists B for this work.

**Competing interests:** The authors have declared that no competing interests exist.

tomography. The As/Ps may provide information for improving the diagnostic accuracy of KC, even in the initial stage of the disease.

## Introduction

Keratoconus (KC) is a progressive corneal disease characterized by local corneal thinning and protrusion of the cornea, thus resulting in loss of vision due to irregular corneal astigmatism [1,2]. It has been reported that the estimated prevalence of KC among the general population is approximately 1 in every 2000 persons, and thus, very common [3]. Following the advent of refractive surgery methods such as photorefractive keratectomy (PRK) and laser-assisted in situ keratomileusis (LASIK), etc., there have been numerous reported cases of myopia progression due to rapid progression of central steepening postsurgery, classified as keratectasia and with a pathology similar to KC [4,5]. Thus, specific diagnostic criteria in regard to the development of KC is required, as it may contribute to not only the prevention of the progression into severe KC, [2,6] but also the overall safety of refractive surgery [5].

KC is basically a bilateral disease [2], and the contralateral cornea of patients with KC without any ectatic changes clinically and topographically has been designated as forme fruste keratoconus (FFKC) [7]. A recent study reported that in 50% of the subjects with unilateral KC, onset of the disease will occur in the non-affected fellow eye within 16 years [8], and that those FFKC eyes were considered to be at the earliest and mildest stage of KC [9]. In previous reports, the corneal surface area showed significant difference between KC and normal eyes [10,11] as well as several diagnostic indices obtained by corneal topography and rotating-Scheimpflug-camera-based corneal tomography that are highly successful for discriminating KC and FFKC from normal healthy eyes [9,12–15]. We previously reported that anterior and posterior corneal-surface-area imbalance derived from areas calculated via the elevation map of anterior-segment optical coherence tomography (AS-OCT) may reflect keratoconic eyes at the early stage of the disease and that using the ratio of anterior-posterior surface area (As/Ps) to distinguish normal eyes from FFKC eyes was 0.948, and quite large [16]. However, and the diagnostic accuracy of As/Ps for discriminating FFKC eyes from normal eyes has not been fully been evaluated and compared to that of other indices.

The purpose of this present study was to investigate the diagnostic ability of As/Ps to detect FFKC by comparing it with the previously-reported KC detection indices provided by AS-OCT and rotating-Scheimpflug-camera-based corneal tomography.

## Methods

### Patient inclusion

In this retrospective case-control study, we examined 13 eyes of 13 patients with FFKC (11 male and, 2 female), 29 eyes of 29 patients with KC (24 male and, 5 female) and 88 eyes of 88 patients with normal subjects (55 male and, 33 female) at Keratoconus and Refractive Subspecialty Clinic in the Baptist Eye Institute, Kyoto, Japan from April 2015 to June 2018. The mean ± standard deviation (SD) ages of patients with FFKC, KC and normal subjects were 28.2 ± 6.4 years (range 16–39 years), 27.1 ± 5.4 years (range 17–39 years), and 30.4 ± 7.3 years (range 16–36 years), respectively. The KC eyes were used to analyze as reference group.

The KC-group eyes were used for reference analysis. The study protocol was approved by the Kyoto Ethics Review Committee, Kyoto, Japan, an independent organization established for the approval of ethics-related issues (Approval #1604). This study was performed in

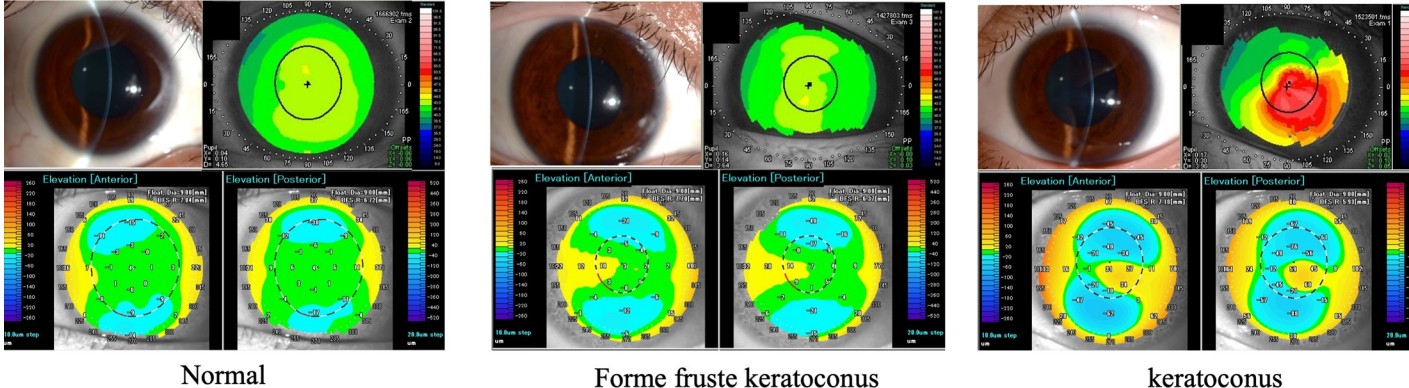

| Normal | Forme fruste keratoconus | keratoconus |

**Fig 1. Representative data of a normal subject, a forme fruste keratoconus patient, and a keratoconus patient.**

accordance with the tenets set forth in the Declaration of Helsinki. Written informed consent was obtained from all subjects and their parents/guardians for patients younger than 20 years old.

Patients were diagnosed as having keratoconus if all the following criteria were found: 1) at least one of the following biomicroscopic signs: corneal Fleischer ring, Vogt's striae, corneal thinning, and corneal protrusion at the apex, 2) keratoconic appearance such as focal or inferior steepening, an asymmetric bow-tie pattern with or without skewed axis [9] on placido-disc based topography (TMS-4, Tomey Corporation, Nagoya, Japan), and 3) the Keratoconus index (KCI) showed 5% or more. The KCI is video-keratographic index for detection keratoconus, calculated by Clyce/Maeda criteria on TMS-4 [13]. Subjects were classified into the FFKC group if 1) they had no clinical and topographic signs of keratoconus in one eye, 2) passed through the TMS-4 keratoconus screening program with 0% KCI, and 3) presented a diagnosis of keratoconus in the contralateral eye. The normal group subjects had no indication of KC, and enrolled as age matched control to FFKC group. Exclusion criteria were the followings: previous ocular surgery, corneal scarring, trauma, acute corneal hydrops, and a history of other ocular disease beside refractive errors. The patient who had not corneal-map with high quality and satisfactory for calculation (refer to Imaging methods) obtained by either AS-OCT or rotating Scheimpflug–based corneal topography was also excluded. Contact lens–wearing patients were asked to stop wearing contact lenses for 4 weeks, in case of rigid contact lenses, and 2 weeks for soft contact lenses, before assessment. Three representative cases are shown in Fig 1.

Anterior segment photo (upper left), Placido-disc based topography (upper right), anterior (bottom left) and posterior (bottom right) corneal elevation map using anterior segment optical coherence tomography (AS-OCT) images in normal subject, forme fruste keratoconus (FFKC) patient and keratoconus (KC) patient.

## Imaging methods

All patients underwent topographic examination via AS-OCT imaging (SS-1000 CASIA; Tomey Corporation, Nagoya, Japan), high-resolution rotating Scheimpflug–based corneal tomography (Pentacam HR, Oculus Optikgerate GmbH, Wetzlar, Germany) and placido-disc based topography (TMS-4, Tomey Corporation, Nagoya, Japan) on the same day. We previously reported that data via AS-OCT has had high reproducibility and repeatability [16].

The AS-OCT is swept-source OCT that uses a center wavelength of 1,310 nm and measures with a speed of 30,000 A lines per second. Using the AS-OCT, 16 cross-sectional images,

which consisted of 512 A-scans were obtained for 0.34 second during 1 measurement to assess the corneal topography. The built-in program identified and digitized the anterior and posterior corneal surfaces for further analysis.

The indices from the AS-OCT included the anterior and posterior mean Ks (steep keratometry), and mean Kf (flat keratometry), the different sector index (DSI), the opposite sector index (OSI), central corneal thickness (CCT) at the apex, corneal thickness at the thinnest point (CTmin), anterior or posterior best-fit sphere (BFS), elevation values, and corneal surface areas, the As/Ps and the ectasia screening index (ESI). The DSI is the greatest difference in the average power between any two sectors and the OSI is the greatest difference in the average power of two opposite sectors from a divided eight pie-shaped image. The BFS was calculated by AS-OCT built-in software with float option within central 9.0 mm cornea and used as reference surface. Anterior or posterior elevations were measured as the maximum value above the BFS in the central 5 mm of the anterior or posterior cornea. The anterior or posterior corneal surfaces were calculated within a central 5.0 mm (not radius) area of cornea based on the anterior or posterior elevation maps, as we previously reported [17]. Briefly, we assumed the cornea was a spherical cap and used the formula $S = 2 \times PI \times R \left\{ R - \sqrt{[R^2 - (D/2)^2]} \right\}$, in which S = surface area, PI = ratio of a circle's circumference (3.14), R = curvature, and D = corneal diameter, at each measurement point based on the elevation map which was calculated by AS-OCT. The calculation scheme is shown in Fig 2.

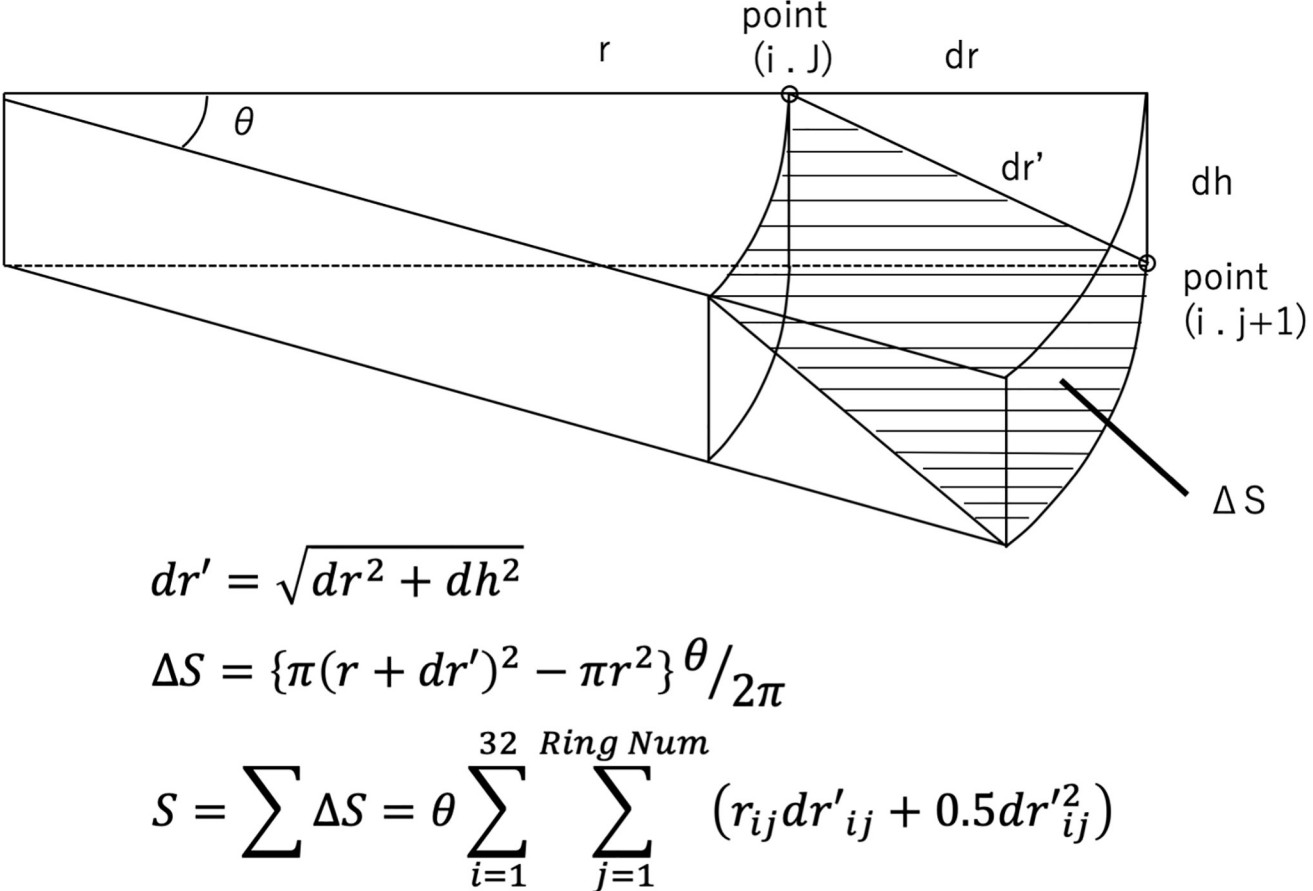

$$dr' = \sqrt{dr^2 + dh^2}$$

$$\Delta S = \{\pi(r + dr')^2 - \pi r^2\} \theta / 2\pi$$

$$S = \sum \Delta S = \theta \sum_{i=1}^{32} \sum_{j=1}^{Ring\ Num} \left( r_{ij} dr'_{ij} + 0.5 dr'^2_{ij} \right)$$

**Fig 2. Schema used to compute the corneal surface area.**

To calculate the corneal surface areas, we assumed the corneal surface as a fan-shaped surface that lacked the central area ($\Delta S$). Each fan-shaped surface was based-on an adjacent point, which was merged into each calculation. The calculation formulas are shown in the top part of the figure, as we previously reported [17]. i = measuring point in the direction of rotation (range: 1–32), j = ring number [1 to RingNum (※)], ※ = outermost ring number (determined by the measurement diameter), r = distance from the center to the measuring point in the XY direction, dr = diameter of the unit segment joining 2 adjacent points [(i, j) and (i, j + 1)], dh = height of the unit segment joining 2 adjacent points [(i, j) and (i, j + 1)], DS = fan-shaped surface lacking the central area.

The As/Ps was calculated as follows: As/Ps = anterior corneal surface area / posterior corneal surface area. The ESI was measured by AS-OCT built-in software which estimates the ectasia similarity of the scan. The ESI consisted of anterior and posterior score derived from Fourier analysis [18]. In each rotating Scheimpflug–based corneal tomography scans measurement, a total of 25 single Scheimpflug images captured within 1 second to measure the anterior and posterior corneal configurations. KC index from the rotating Scheimpflug-based corneal tomography included the index of surface variance (ISV), the index of vertical asymmetry (IVA), the maximum (PPImax), minimum (PPImin) and the average (PPIave) pachymetric progression index, the maximum Ambrosio relational thickness (ARTmax), the back difference elevation (BDE) and the Belin/Ambrosio display enhanced ectasia total derivation value (BAD-D). The PPI was calculated as the progression value at the different rings referenced to the mean curve. The ARTmax was calculated as follows: ARTmax = the corneal thickness at the thinnest point / PPImax. The BDE and BAD-D were extrapolated from the difference map of the Belin/Ambrosio-enhanced ectasia display.

Placido-disc based topography takes 4 measurements capturing images of 25 mire ring during 3.0 seconds, and evaluates 256 points on each 25 mire rings. We evaluated Keratoconus Index which is a video-keratographic index calculated by Klyce/Maeda criteria on TMS-4, which is based on linear discriminant analysis and a binary decision tree [14].

In each examined eye, a corneal map with at least 9.0 mm of corneal coverage and no extrapolated data by AS-OCT and rotating Scheimpflug corneal tomography respectively, was required.

## Statistical analyses

Statistical analyses were accomplished using R version 3.1.0 (The R Foundation) statistical software, with the data being presented as mean ± SD where applicable. The data were not normally distributed; therefore, Kruskal-Wallis test followed by paired Wilcoxon test was used to compare each parameter between normal eyes and FFKC or KC eyes. A Bonferroni correction was used to control type I error. A P-value less than 0.05 was considered to be statistically significant. Receiver- operating-characteristic (ROC) curves were used to determine the overall predictive accuracy of the test parameters as described by the area under the curve (AUC) and to calculate the sensitivity and specificity in distinguishing FFKC from normal eyes. Optimal cutoff points for each index were obtained from the ROC curves as those closest to the particular point, where sensitivity equals specificity. This point of the curve is where the product of these two indices (ie. sensitivity and specificity) is maximum. [19]

## Results

### Intergroup differences

The median of all parameters used for statistical analysis in all groups was summarized in Table 1. Statistically significant differences were noted between normal and FFKC eyes for all

**Table 1. All parameters of Normal, FFKC and KC eyes acquired from AS-OCT and rotating Scheimpflug tomography.**

| Indices, Mean ± SD | | Normal (n = 88) | FFKC (n = 13) | KC (n = 29) | N vs FFKC | N vs KC |
|---|---|---|---|---|---|---|
| **AS-OCT** | aKs | 48.62 ± 1.52 | 48.84 ± 2.74 | 60.73 ± 8.09 | 0.33 | p < 0.01 |
| | aKf | 47.31 ± 1.36 | 46.69 ± 2.32 | 56.27 ± 7.25 | 0.52 | p < 0.01 |
| | pKs | -6.30 ± 0.23 | -6.40 ± 0.36 | -7.78 ± 1.87 | 0.23 | p < 0.01 |
| | pKf | -5.98 ± 0.18 | -6.04 ± 0.20 | -7.33 ± 1.51 | 0.48 | p < 0.01 |
| | DSI | 1.61 ± 0.60 | 2.87 ± 1.13 | 11.27 ± 4.84 | p < 0.01 | p < 0.01 |
| | OSI | 0.77 ± 0.46 | 1.98 ± 1.07 | 10.69 ± 4.80 | p < 0.01 | p < 0.01 |
| | CCT | 541.15 ± 26.23 | 501.46 ± 34.83 | 438.07 ± 60.87 | p < 0.01 | p < 0.01 |
| | CTmin | 536.86 ± 25.96 | 493.46 ± 36.88 | 411.86 ± 59.03 | p < 0.01 | p < 0.01 |
| | aBFS | 7.88 ± 0.20 | 7.88 ± 0.45 | 6.45 ± 0.89 | 0.31 | p < 0.01 |
| | pBFS | 6.57 ± 0.23 | 6.48 ± 0.32 | 5.10 ± 0.93 | 0.18 | p < 0.01 |
| | As | 20.15 ± 0.03 | 20.14 ± 0.06 | 20.40 ± 0.20 | 0.63 | p < 0.01 |
| | Ps | 20.38 ± 0.05 | 20.44 ± 0.07 | 20.90 ± 0.36 | p < 0.01 | p < 0.01 |
| | As/Ps | 0.989 ± 0.001 | 0.986 ± 0.001 | 0.976 ± 0.008 | p < 0.01 | p < 0.01 |
| | AE | 5.43 ± 1.38 | 13.85 ± 7.23 | 44.00 ± 22.15 | p < 0.01 | p < 0.01 |
| | PE | 10.00 ± 2.82 | 22.69 ± 12.07 | 89.93 ± 41.93 | p < 0.01 | p < 0.01 |
| | ESI | 0.00 ± 0.00 | 5.77 ± 6.98 | 76.00 ± 24.88 | p < 0.01 | p < 0.01 |
| **Rotating Scheimpflug tomography** | ISV | 16.19 ± 5.67 | 27.85 ± 13.99 | 109.69 ± 48.68 | p < 0.01 | p < 0.01 |
| | IVA | 0.10 ± 0.04 | 0.26 ± 0.19 | 1.05 ± 0.40 | p < 0.01 | p < 0.01 |
| | PPImin | 0.71 ± 0.14 | 0.89 ± 0.26 | 2.29± 1.24 | p < 0.01 | p < 0.01 |
| | PPImax | 1.28 ± 0.18 | 1.67 ± 0.43 | 4.42± 1.85 | p < 0.01 | p < 0.01 |
| | PPIave | 1.00 ± 0.16 | 1.31 ± 0.24 | 2.98 ± 1.33 | p < 0.01 | p < 0.01 |
| | BDE | 5.31 ± 3.40 | 8.92 ± 4.29 | 55.83 ± 36.95 | p < 0.01 | p < 0.01 |
| | ARTmax | 436.30 ± 73.69 | 318.08 ± 84.24 | 118.07± 57.10 | p < 0.01 | p < 0.01 |
| | BAD-D | 0.94 ± 0.48 | 2.11 ± 0.77 | 12.34 ± 6.28 | p < 0.01 | p < 0.01 |

FFKC: forme fruste keratoconus, KC: keratoconus, N: Normal, aKs: Anterior steep keratometry, aKf: Anterior flat keratometry, pKs: Posterior steep keratometry, pKf: Posterior flat keratometry, DSI: differential sector index, OSI: Opposite sector index, CCT: central corneal thickness, CTmin: corneal thickness at the thinnest point, aBFS: anterior best-fit sphere, pBFS: posterior best-fit sphere, As: anterior corneal surface area, Ps: posterior corneal surface area, As/Ps: ratio of anterior and posterior corneal surface area, AE: anterior elevation values, PE: posterior elevation values ESI: ectasia screening index, ISV: index of surface variance, IVA: index of vertical asymmetry, PPImin: minimum pachymetric progression index, PPImax: maximum pachymetric progression index, PPIave: average pachymetric progression index, BDE: back difference elevation, ARTmax: maximum Ambrosio relation thickness, BAD-D: Belin/Ambrosio display enhanced ectasia total derivation value, SD: standard deviation.

parameters with the exception of anterior and posterior keratometric reading (i.e., Ks and Kf), BFS and anterior surface areas.

To evaluate the ability of indices to discriminate between normal eyes and FFKC eyes, the results of the ROC analysis including AU-ROC and 95% confidence intervals (CI) for all parameters were plotted (Table 2). The AU-ROC of the As/Ps, the Belin/Ambrosio display enhanced ectasia total derivation value (BAD-D) and posterior and anterior elevation were 0.9 or more in differentiating FFKC from normal eyes (0.980, 0.951, 0.924 and 0.903, respectively). The cut-off points with sensitivity and specificity of As/Ps were 0.98 with 0.92 and 0.96, respectively. The sensitivity and specificity of other indices were 1.00 and 0.90 for BAD-D, 0.85 and 0.86 for posterior elevation value, and 0.85 and 0.96 for anterior elevation value, respectively.

The ROC curves of these parameters to differentiate FFKC from normal eyes are shown in Fig 3.

Combined receiver operating characteristic curves for the ratio of anterior and posterior corneal surface areas (As/Ps), Belin/Ambrosio display enhanced ectasia total derivation value

**Table 2.  Receiver-operating-characteristic curve analysis for the Normal versus FFKC eyes.**

| | Indices | AU-ROC | 95% CI | Cut-off | Sensitivity | Specificity |
|---|---|---|---|---|---|---|
| **AS-OCT** | aKs | 0.584 | 0.374–0.795 | 50.26 | 0.46 | 0.84 |
| | aKf | 0.556 | 0.363–0.750 | 47.00 | 0.54 | 0.59 |
| | pKs | 0.603 | 0.396–0.810 | -6.42 | 0.62 | 0.66 |
| | pKf | 0.566 | 0.367–0.765 | -6.15 | 0.46 | 0.76 |
| | DSI | 0.826 | 0.667–0.984 | 1.95 | 0.85 | 0.76 |
| | OSI | 0.868 | 0.737–0.999 | 1.24 | 0.85 | 0.86 |
| | CCT | 0.809 | 0.648–0.971 | 507.00 | 0.69 | 0.89 |
| | CTmin | 0.822 | 0.665–0.980 | 500.00 | 0.69 | 0.91 |
| | aBFS | 0.587 | 0.353–0.821 | 7.72 | 0.54 | 0.78 |
| | pBFS | 0.615 | 0.413–0.818 | 6.48 | 0.62 | 0.59 |
| | As | 0.542 | 0.333–0.751 | 20.20 | 0.46 | 0.71 |
| | Ps | 0.751 | 0.548–0.955 | 20.42 | 0.69 | 0.80 |
| | As/Ps | 0.98 | 0.954–1.00 | 0.99 | 0.92 | 0.96 |
| | AE | 0.903 | 0.770–1.000 | 8.00 | 0.85 | 0.96 |
| | PE | 0.924 | 0.853–0.997 | 13.00 | 0.85 | 0.86 |
| | ESI | 0.731 | 0.59–0.872 | 7.00 | 1.00 | 0.46 |
| **Rotating Scheimpflug tomography** | ISV | 0.801 | 0.659–0.943 | 22.00 | 0.69 | 0.83 |
| | IVA | 0.884 | 0.75–1.000 | 0.16 | 0.85 | 0.85 |
| | PPImin | 0.782 | 0.606–0.958 | 0.85 | 0.69 | 0.84 |
| | PPImax | 0.832 | 0.692–0.973 | 1.37 | 0.85 | 0.78 |
| | PPIave | 0.866 | 0.74–0.993 | 1.15 | 0.77 | 0.85 |
| | BDE | 0.743 | 0.583–0.904 | 7.00 | 0.69 | 0.67 |
| | ARTmax | 0.871 | 0.749–0.992 | 388.00 | 0.92 | 0.78 |
| | BAD-D | 0.951 | 0.910–0.991 | 1.33 | 1.00 | 0.90 |

AU-ROC: the area under the receiver operation characteristics curve, CI: cofidence interval, aKs: Anterior steep keratometry, aKf: Anterior flat keratometry, pKs: Posterior steep keratometry, pKf: Posterior flat keratometry, DSI: differential sector index, OSI: Opposite sector index, CCT: central corneal thickness, CTmin: corneal thickness at the thinnest point, aBFS: anterior best-fit sphere, pBFS: posterior best-fit sphere, As: anterior corneal surface area, Ps: posterior corneal surface area, As/Ps: ratio of anterior and posterior corneal surface area, AE: anterior elevation values, PE: posterior elevation values ESI: ectasia screening index, ISV: index of surface variance, IVA: index of vertical asymmetry, PPImin: minimum pachymetric progression index, PPImax: maximum pachymetric progression index, PPIave: average pachymetric progression index, BDE: back difference elevation, ARTmax: maximum Ambrosio relation thickness, BAD-D: Belin/Ambrosio display enhanced ectasia total derivation value.

(BAD-D), Posterior elevation values (PE) to differentiate forme fruste keratoconus (FFKC) from normal controls. Note that the As/Ps has the largest AUC.

## Discussion

In this present study, we found that the As/Ps calculated by elevation map of the AS-OCT, which we previously reported as a novel comprehensive index reflecting the both of anterior and posterior cornea surfaces, revealed the large AU-ROC (0.980). Although previous geometric reports demonstrated anterior surface and posterior surface areas as new indices [10,11], to the best our knowledge, the present study is a first report showing the discriminating ability of the As/Ps obtained by AS-OCT to detect FFKC via comparison with several KC screening parameters obtained by AS-OCT and rotating Scheimpflug–based corneal tomography.

In order to detect early stage of KC, there were already a great variety of indices, derived from the posterior elevation parameters, corneal thickness and corneal aberration [9,12,13,20]. In the studies of index measured by Scheimpflug imaging system, the ART max [12], the MAX

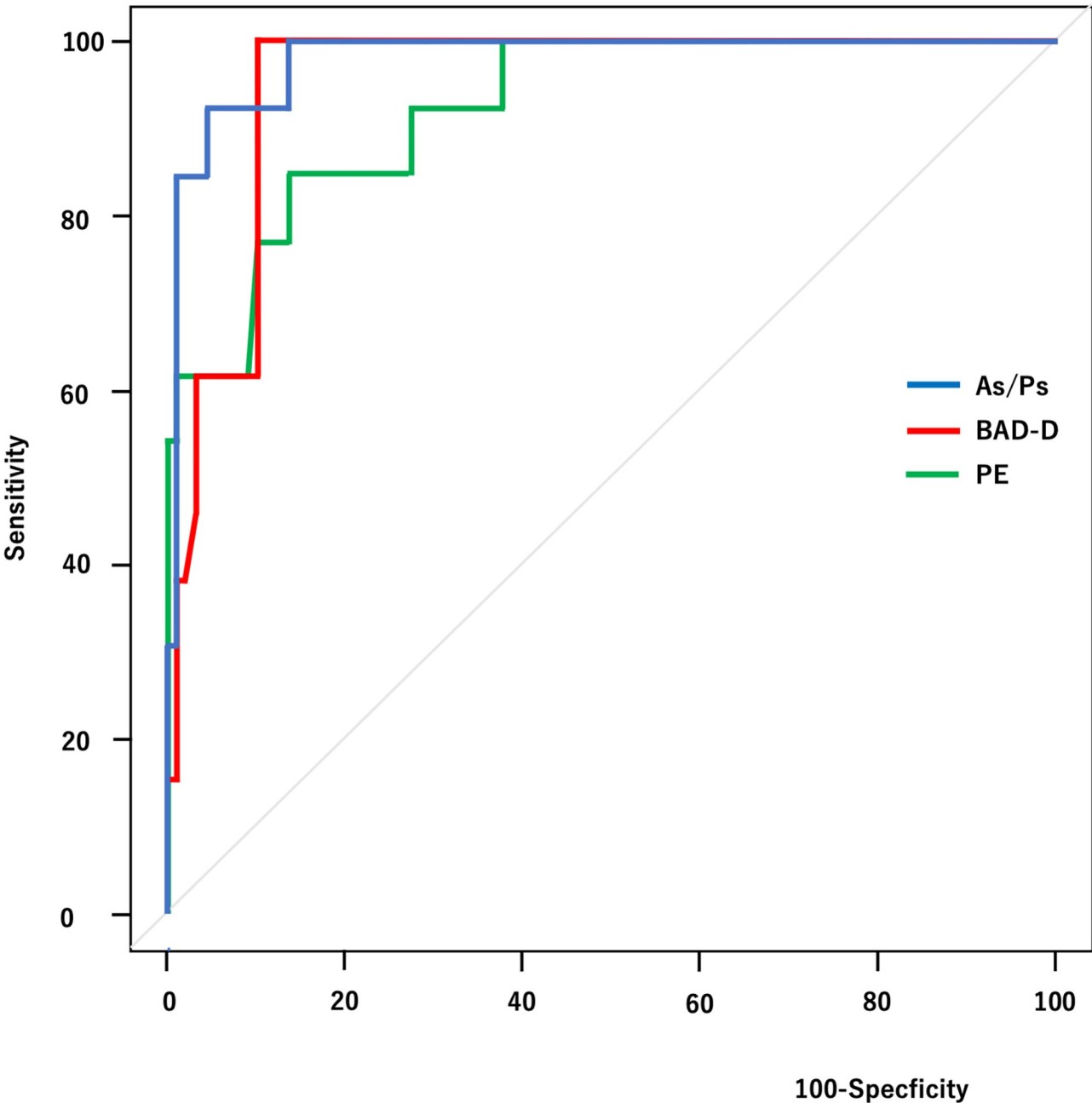

**Fig 3. Receiver operating characteristic curves of each index.**

PE5 [13], which measured the maximum posterior elevation from the reference BFS at the central 5.0 mm of cornea, and the back difference elevation (BDE) [9] were reported to show the large AU-ROC in FFKC compared to normal eyes as 0.88 (0.82 sensitivity and 0.70 specificity), 0.93 (0.68 sensitivity and 0.70 specificity)and 0.76 (0.74 sensitivity and 0.65 specificity), respectively. Naderan and associates [20] found that the ocular total higher aberration had the highest diagnostic ability in discriminating FFKC from normal eyes, as 0.946 (0.94 sensitivity and 0.94 specificity). However, those previous studies were comprised of a different patient

population with different measurement techniques. Thus, those indices cannot be directly compared with the diagnostic accuracy of the As/Ps in this present study.

The corneal surface area ratio in this preset study used data at the central 5.0-mm, because our previous study revealed that AU-ROC of As/Ps at the central 5.0-mm areas was larger than that at the central 6.0-mm and 7.0-mm.15 This might possibly be explained by that the apex of the both anterior and posterior surfaces was located within the central 5.0-mm, and the difference between anterior surface are and posterior surface at larger area was underestimated.

In this present study, AU-ROC of the As/Ps, BAD-D and anterior and posterior elevation values showed 0.9 or more, and furthermore, the As/Ps and BAD-D showed high sensitivity and specificity (>0.9). The test with AU-ROC greater than 0.9 can be interpreted as having high accuracy [21], thus our results suggest that the As/Ps had sufficient strength in differentiate FFKC from normal eyes. Although this finding does not reduce the importance of known indices which showed AU-ROC of less than 0.9, it highlights the importance of the As/Ps derived from AS-OCT in detecting KC at the early stage of disease.

In this present study, the As/Ps had the large AU-ROC (0.980), in distinguishing the FFKC from the normal eyes. This large AU-ROC in As/Ps was explained by the characteristic of the index, which included not only the components of anterior and posterior cornea surfaces, but also imbalance of anterior-posterior ratio, and corneal thickness, as we previously reported [16]. However, there is ongoing debates as to whether changes in the anterior surface [22], posterior surface [2,9], or both surface of the cornea [23] are the first arise in the early stage of KC. Our findings, which demonstrated the smaller As/Ps in FFKC and KC eyes, suggest that protrusion of the posterior corneal surface prior to changes being observed on the anterior corneal surface, leading to be imbalance of anterior and posterior corneal surface area at the initial stage of the disease. The BAD-D, also provided the large AU-ROC (0.951) in our study, and it has been reported as multi-metric parameter, with calculation based on the front and back enhanced elevations, thinnest value, pachymetric distribution, and vertical displacement of the thinnest in relation to the apex [24]. In previous reports, the combined index generated from the curvature, elevation, and corneal thickness has showed high accuracy in detecting subclinical keratoconus or forme fruste keratoconus [7,25]. The findings in those studies may support our finding in which several components index, namely the As/Ps and BAD-D, had higher diagnostic ability than single component index in early-stage KC and indicate the importance of comprehensive approach to characterize the initial change in KC. Consequently, the character of the As/Ps as multi-metric index which reflects the imbalance of anterior-posterior ratio, leads to large AU-ROC in distinguishing FFKC from normal eyes.

It should be noted that this current study did have several limitations. First, in order to be widely applied the As/Ps for the screening of the early stage of KC, the optimal cut-off value should be determined. This was a retrospective study and it involved a small number of cases because most of subjects were not able to stop wearing the lenses for 2 to 4 weeks prior to assessment, and did not have the images with high quality from both AS-OCT and Scheimpflug imaging system. Hence, we are now conducting the extensive research to determine the optimal cut-off value using another patient population in a larger sample size. Second, we did not evaluate the corneal biomechanics or the total corneal wavefront aberrations, which are reportedly useful in the diagnosis of ectatic disease [19, 26–28]. Thus, in order to identify the significant independent predictors of the presence of early-stage KC, additional investigation involving logistic regression analysis with a large number of cases is required. We hope that the present findings lead to the early detection of KC, and then the early therapeutic intervention including cross-linking will be administered for preventing progress to severe KC.

In conclusion, it was found that the As/Ps, which was calculated by using the surface area obtained from AS-OCT, was a reliable index among the several previously reported indices, demonstrating the large AU-ROC (0.980) in detecting FFKC.

## Supporting information

**S1 Dataset. Data analyzed.**
(XLSX)

## Acknowledgments

The authors wish to thank John Bush for reviewing the manuscript.

## Author Contributions

**Conceptualization:** Koji Kitazawa.

**Data curation:** Motohiro Itoi, Koji Kitazawa, Isao Yokota.

**Formal analysis:** Motohiro Itoi.

**Funding acquisition:** Koji Kitazawa.

**Investigation:** Motohiro Itoi, Koji Kitazawa, Isao Yokota.

**Methodology:** Motohiro Itoi, Koji Kitazawa, Isao Yokota, Satoshi Teramukai.

**Project administration:** Motohiro Itoi, Koji Kitazawa.

**Resources:** Motohiro Itoi, Koji Kitazawa, Isao Yokota, Koichi Wakimasu, Yuko Cho, Yo Nakamura, Osamu Hieda, Satoshi Teramukai, Shigeru Kinoshita, Chie Sotozono.

**Software:** Motohiro Itoi.

**Supervision:** Koji Kitazawa, Yo Nakamura, Osamu Hieda, Satoshi Teramukai, Chie Sotozono.

**Validation:** Motohiro Itoi.

**Visualization:** Motohiro Itoi.

**Writing – original draft:** Motohiro Itoi, Koji Kitazawa.

**Writing – review & editing:** Motohiro Itoi, Koji Kitazawa, Chie Sotozono.

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
