## [Decision Letter · Decision Letter 0]

6 Jan 2020

PONE-D-19-34557

Anterior and Posterior Ratio of Corneal Surface Areas: A Novel Index for Detecting Early Stage Keratoconus

PLOS ONE

Dear Dr. Kitazawa,

Thank you for submitting your manuscript to PLOS ONE. After careful consideration, we feel that it has merit but does not fully meet PLOS ONE’s publication criteria as it currently stands. Therefore, we invite you to submit a revised version of the manuscript that addresses the points raised during the review process.

We would appreciate receiving your revised manuscript by Feb 20 2020 11:59PM. To enhance the reproducibility of your results, we recommend that if applicable you deposit your laboratory protocols in protocols.io, where a protocol can be assigned its own identifier (DOI) such that it can be cited independently in the future. For instructions see: http://journals.plos.org/plosone/s/submission-guidelines#loc-laboratory-protocols

We look forward to receiving your revised manuscript.

Kind regards,

Yu-Chi Liu, M.D

Academic Editor

PLOS ONE

Journal Requirements:

2. Thank you for your financial disclosure: "Koji Kitazawa received Grant-in-Aid for Young Scientists B for this work."

Please provide, in the cover letter accompanying your revised manuscript a  reworded financial disclosure which states specifically whether the funders played any role in the study.

3. Please carefully proofread your manuscript for typographical errors. For example, in the Patient Inclusion section there is a missing space “… 88 eyes of 88patients ….”.

4. Please state the participant recruitment date in your methods section.

5. You indicated that you had ethical approval for your study. In your Methods section, please ensure you have also stated whether you obtained consent from parents or guardians of the minors included in the study or whether the research ethics committee or IRB specifically waived the need for their consent.

Reviewers' comments:

Reviewer's Responses to Questions

**Comments to the Author**

1. Is the manuscript technically sound, and do the data support the conclusions?

Reviewer #1: Yes

Reviewer #2: Partly

Reviewer #3: Yes

2. Has the statistical analysis been performed appropriately and rigorously? 

Reviewer #1: Yes

Reviewer #2: No

Reviewer #3: Yes

3. Have the authors made all data underlying the findings in their manuscript fully available?

Reviewer #1: Yes

Reviewer #2: No

Reviewer #3: Yes

4. Is the manuscript presented in an intelligible fashion and written in standard English?

Reviewer #1: Yes

Reviewer #2: Yes

Reviewer #3: No

5. Review Comments to the Author

Reviewer #1: it is a novel work and should be studied on larger population and other ethnic groups to be applicable and dependable anywhere. minimal writing errors need correction. I recommend accepting this manuscript.

Reviewer #2: There are some important points I would like to address:

Methods: Lines 126- 133

1. Explain KCN and FFKCN classification methods more clearly. For example, how much skewed-axis was included in the KCN group? Same for other group classifications.

Analysis:

2. The important point in this paper is this study does not evaluate the sensitivity and specificity of the As/Ps index compared to other indices. If you want to say that this index is better than the others, a specific analysis should be performed to compare the area under the curves (AUCs). The area under the curve alone cannot confirm As/Ps ratio has a high diagnostic ability for early KCN detection. It is recommended to use the “DeLong” method to compare the area under the curves and report sensitivity, specificity and optimal cut points. In retrospective data reporting these diagnostic indices also possible.

Results:

3. Line 117, you said keratosis was a reference for the analysis. Why did you compare FFKCN with the control group in Table 1?

4. You should remove the KCN column from the table or report that the p-value related to the comparison of KCN with the normal group.

Discussion:

5. Explain in the discussion why the AUC of As had weak value however, the anterior (As) and posterior (Ps) ratio had high AUC value. (based on Table 2)

6. Curvature indices (e.g ISValue) and new combined indices (e.g BAD-D) appear to had better diagnostic accuracy for early KCN detection. Why this new elevation based index (As/Ps ) had better diagnostic power than curvature indices OR new combined indices to detect early stages of the disease?

Reviewer #3: The authors have retrospectively collected 13 forme fruste keratoconus (FFKC) eyes, 29 keratoconus eyes, and 88 normal eyes and analyze their keratoconus related indices by three devices: anterior segment OCT (CASIA), Scheimpflug imaging tomography (Pentacam), and Placido disc-based topography (TMS-4). In their cohort, they reported the ratio of anterior/posterior corneal surfaces (As/Ps)(CASIA) has the largest AUROC in differentiating FFKC from normal eyes, which may improve the diagnosis of early keratoconus. This study is an extended study of the authors' previous work (Ref 15) that focused on the imbalance of anterior/posterior corneal surface as a pathological change of keratoconus. The experiments, statistics and data analyses are performed in an adequate technical standard.

Few specific suggestions are listed below:

Material:

1. Line 124-125: (2) keratoconic appearance such as focal or inferior steepening, a “bow-tie pattern, or a skewed axis.” Should the author clarify the criteria as “an asymmetric bow-tie pattern with or without skewed axis?” (Ref: 8, Muftuoglu O, et al. J Cataract Refract Surg 2013)

2. Line 172: formula for calculation the posterior corneal surface is a bit confusing. Please explain the assumption and derivation of your formula and present it in a clear way with references.

3. Line 176: please explain the derivation of ESI.

General recommendation:

1. Use Scheimpflug tomography for Pentacam instead of topography since the rotating Scheimpflug imaging system also provides posterior corneal parameters and pachymetry data.

2. Please check all the references in your manuscript, some references are missing.

6. PLOS authors have the option to publish the peer review history of their article (what does this mean?). If published, this will include your full peer review and any attached files.

Reviewer #1: Yes: Ismail Ahmed Nagib Omar

Reviewer #2: No

Reviewer #3: No

---

## [Author Response · Author response to Decision Letter 0]

25 Jan 2020

Response letter

Response to Comment 1: We greatly appreciate the Editor’s very helpful comment. We have revised the manuscripts to meet PLOS ONE’s style requirements throughout the paper. (i.e. Heading with Bold type, Figure citation, Inline Equation, Reference Citation)

2. Thank you for your financial disclosure: "Koji Kitazawa received Grant-in-Aid for Young Scientists B for this work." Please provide, in the cover letter accompanying your revised manuscript a reworded financial disclosure which states specifically whether the funders played any role in the study.

Response to Comment 2: We greatly appreciate the Editor’s very insightful and helpful comment. To clarify, we have added the text regarding the financial disclosure in the cover letter.

3. Please carefully proofread your manuscript for typographical errors. For example, in the Patient Inclusion section there is a missing space “… 88 eyes of 88patients ….”. 

Response to Comment 3: We greatly appreciate the Editor’s very insightful and helpful comment. We have revised the manuscripts in regard to the typographical errors throughout the paper.

4. Please state the participant recruitment date in your methods section.

Response to Comment 4: We greatly appreciate the Editor’s very insightful and helpful comment. As suggested, please note that we have now added the text regarding the participant recruitment date in method section. (p. 6, lines 2-3)

5. You indicated that you had ethical approval for your study. In your Methods section, please ensure you have also stated whether you obtained consent from parents or guardians of the minors included in the study or whether the research ethics committee or IRB specifically waived the need for their consent.

Response to Comment 5: We greatly appreciate the Reviewer’s very insightful and helpful comment. We have now added the text regarding informed consent in the Method sections. (p. 6, lines 11-12)

Response to Comment 6: We greatly appreciate the Editor’s very insightful and helpful comment. We have now added the standard deviations of all indices which were reported in our study in table 1. Since the values behind the means, sample size, statistical method, P value, the values used for ROC analysis in figure 3 were included in our manuscripts. We believe that this new information adequately addresses the editor's comment.

Review Comments to the Author

Reviewer #1: it is a novel work and should be studied on larger population and other ethnic groups to be applicable and dependable anywhere. minimal writing errors need correction. I recommend accepting this manuscript.

Response to Comment1: We greatly appreciate the Reviewer’s very helpful comment. As suggested, please note that we have now added the standard deviation of each indices to correct minimal writing errors in table 1.

Reviewer #2: There are some important points I would like to address:

Methods: Lines 126- 133

1. Explain KCN and FFKCN classification methods more clearly. For example, how much skewed-axis was included in the KCN group? Same for other group classifications.

Response to Comment 1: We greatly appreciate the Reviewer’s very insightful and helpful comment. We used the Keratoconus index (KCI), calculated by Clyce/Maeda criteria on TMS-4, as objective indication for diagnosis of KC and FFKC. Patients were diagnosed as having keratoconus if they have clinical and topographic signs and showed KCI 5 % or more. All patients enrolled in the FFKC group had to pass through the KCI with 0%. We have now added some text in the Method sections to describe the classification methods for KC and FFKC. (p. 6, lines 12-18 and p. 7, lines 1-4)

Analysis:

2. The important point in this paper is this study does not evaluate the sensitivity and specificity of the As/Ps index compared to other indices. If you want to say that this index is better than the others, a specific analysis should be performed to compare the area under the curves (AUCs). The area under the curve alone cannot confirm As/Ps ratio has a high diagnostic ability for early KCN detection. It is recommended to use the “DeLong” method to compare the area under the curves and report sensitivity, specificity and optimal cut points. In retrospective data reporting these diagnostic indices also possible.

Response to Comment 2: We greatly appreciate the Reviewer’s very insightful and helpful comment.　We consulted another biostatistics expert to clarify the method to evaluate diagnostic ability of the As/Ps. We received his opinion as below ”It is difficult to compare and judge the diagnostic accuracy of each indices using sensitivity and specificity in this study. Because the indices which were used in this paper, were not defined universal cut-off value to distinct FFKC from normal subjects. As sensitivity and specificity of each indices changed depending on cut-off value, it is difficult to use sensitivity and specificity in judging diagnostic accuracy of indices which were not defined universal cut-off value. In addition, it is also difficult to compare the AU-ROC via “Delong method” in this study, because it is not easy to find the difference in AUROC using multiple comparison analysis among the numerous indices.” In the other word, it is difficult to　conclude superiority of the As/Ps in diagnostic accuracy. Thus, we tone down our statements and added the text regarding to this matter in Discussion section. We revised “largest” AU-ROC to “large” AU-ROC in order to avoid misleading. We believe that this new statement adequately addresses the reviewer's comment. (p. 2, line 15, p. 3, line 3, p. 18, lines 15-17, p. 20, line 2 and p. 21, line 2)

Results:

3. Line 117, you said keratosis was a reference for the analysis. Why did you compare FFKCN with the control group in Table 1?

Response to Comment 3: We greatly appreciate the Reviewer’s very insightful and helpful comment. The aim of this study was to evaluate the diagnostic ability of the ratio of anterior and posterior corneal surface areas (As/Ps) comparing with other keratoconus screening indices in distinguishing forme fruste keratoconus (FFKC) from normal eyes. We removed the text and add the text (p. 12, line 4). We hope these revisions clarify the aim of our study.

4. You should remove the KCN column from the table or report that the p-value related to the comparison of KCN with the normal group.

Response to Comment 4: We greatly appreciate the Reviewer’s very insightful and helpful comment.　In accordance with the Reviewer’s request, please note that we have now added the p-value related to the comparison of KC with the normal group in table 1, as well as a description of appropriate statistical analysis in the main text. (p. 11, lines 16-18 and p. 12, line 1)

Discussion:

5. Explain in the discussion why the AUC of As had weak value however, the anterior (As) and posterior (Ps) ratio had high AUC value. (based on Table 2)

Response to Comment 5: We greatly appreciate the Reviewer’s very insightful and helpful comment. We think that protrusion of the posterior corneal surface prior to changes being observed on the anterior corneal surface lead to small AU-ROC in As, while As/Ps showing large AU-ROC. Although, there have been numerous previous reports in regard to either anterior or posterior curvature in formation of KC eyes, the findings in our study suggest that the changes in the posterior cornea occurred earlier than the anterior cornea. We have now added some text in the Discussion sections to describe the reason why the AUC of As had weak value however, the anterior (As) and posterior (Ps) ratio had high AUC value. (p. 19, lines 4-10)

6. Curvature indices (e.g ISValue) and new combined indices (e.g BAD-D) appear to had better diagnostic accuracy for early KCN detection. Why this new elevation based index (As/Ps ) had better diagnostic power than curvature indices OR new combined indices to detect early stages of the disease?

Response to Comment 6: We greatly appreciate the Reviewer’s very insightful and helpful comment. As suggested, there has been numerous previous reports in regard to the diagnostic ability of curvature indices and new combined indices in detecting KC. We do not intend to reduce the importance of these known indices. We have now removed the text to avoid misleading and added the text to clarify the points we attempted to make in Discussion section. (p. 2, line 15, p. 3, line 3, p. 18, lines 15-17, p. 20, line 2 and p. 21, line 2)

Reviewer #3: The authors have retrospectively collected 13 forme fruste keratoconus (FFKC) eyes, 29 keratoconus eyes, and 88 normal eyes and analyze their keratoconus related indices by three devices: anterior segment OCT (CASIA), Scheimpflug imaging tomography (Pentacam), and Placido disc-based topography (TMS-4). In their cohort, they reported the ratio of anterior/posterior corneal surfaces (As/Ps)(CASIA) has the largest AUROC in differentiating FFKC from normal eyes, which may improve the diagnosis of early keratoconus. This study is an extended study of the authors' previous work (Ref 15) that focused on the imbalance of anterior/posterior corneal surface as a pathological change of keratoconus. The experiments, statistics and data analyses are performed in an adequate technical standard.

Few specific suggestions are listed below:

Material:

1. Line 124-125: (2) keratoconic appearance such as focal or inferior steepening, a “bow-tie pattern, or a skewed axis.” Should the author clarify the criteria as “an asymmetric bow-tie pattern with or without skewed axis?” (Ref: 8, Muftuoglu O, et al. J Cataract Refract Surg 2013)

Response to Comment 1: We greatly appreciate the Reviewer’s very insightful and helpful comment. We revised the text in the Method section to clarify the criteria, which we used for diagnosis of KC. (p. 6, lines 14-16) 

2. Line 172: formula for calculation the posterior corneal surface is a bit confusing. Please explain the assumption and derivation of your formula and present it in a clear way with references.

Response to Comment 2: We greatly appreciate the Reviewer’s very insightful and helpful comment. In accordance with the Reviewer’s request, please note that we have now added the new figure and text regarding the assumption and derivation of formula with reference citation in Method section. (p. 9, lines 9-10 and 13 and Figure 2)

3. Line 176: please explain the derivation of ESI.

Response to Comment 3: We greatly appreciate the Reviewer’s very insightful and helpful comment. As suggested, please note that that we have now added appropriate reference citation in regards to the derivation of ESI in the Method section. (p. 10, lines 9-12)

General recommendation:

1. Use Scheimpflug tomography for Pentacam instead of topography since the rotating Scheimpflug imaging system also provides posterior corneal parameters and pachymetry data.

Response to Comment 4: We greatly appreciate the Reviewer’s very insightful and helpful comment. We have replaced the term Scheimpflug topography for Pentacam with Scheimpflug tomography throughout the paper to use more precise terms.

2. Please check all the references in your manuscript, some references are missing.

Response to Comment 5: We greatly appreciate the Reviewer’s very insightful and helpful comment. We have reflected this comment and now revised the text in Reference section.

---

## [Decision Letter · Decision Letter 1]

19 Feb 2020

PONE-D-19-34557R1

Anterior and posterior ratio of corneal surface areas: A novel index for detecting early stage keratoconus

PLOS ONE

Dear Dr. Kitazawa,

Thank you for submitting your manuscript to PLOS ONE. The authors have addressed the majority of comments, however, few minor points still need to be revised.  We invite you to submit a revised version of the manuscript that addresses the points raised during the review process.

We would appreciate receiving your revised manuscript by Apr 04 2020 11:59PM. To enhance the reproducibility of your results, we recommend that if applicable you deposit your laboratory protocols in protocols.io, where a protocol can be assigned its own identifier (DOI) such that it can be cited independently in the future. For instructions see: http://journals.plos.org/plosone/s/submission-guidelines#loc-laboratory-protocols

We look forward to receiving your revised manuscript.

Kind regards,

Yu-Chi Liu, M.D

Academic Editor

PLOS ONE

Reviewers' comments:

Reviewer's Responses to Questions

**Comments to the Author**

1. If the authors have adequately addressed your comments raised in a previous round of review and you feel that this manuscript is now acceptable for publication, you may indicate that here to bypass the “Comments to the Author” section, enter your conflict of interest statement in the “Confidential to Editor” section, and submit your "Accept" recommendation.

Reviewer #2: (No Response)

Reviewer #3: (No Response)

2. Is the manuscript technically sound, and do the data support the conclusions?

Reviewer #2: Partly

Reviewer #3: Yes

3. Has the statistical analysis been performed appropriately and rigorously? 

Reviewer #2: No

Reviewer #3: Yes

4. Have the authors made all data underlying the findings in their manuscript fully available?

Reviewer #2: Yes

Reviewer #3: No

5. Is the manuscript presented in an intelligible fashion and written in standard English?

Reviewer #2: Yes

Reviewer #3: Yes

6. Review Comments to the Author

Reviewer #2: Comment 2 (analysis section) not been addressed correctly. You can select the optimal cut-off points based on lowest difference between specificity and sensitivity for your cases. (with SPSS or STATA software)

The "larger AUC" is not correct for comparing the variables (in the abstract and text).

Table 2 should be revised accordingly.

Reviewer #3: The authors have revised the majority of the manuscript according to the reviewers' suggestions. However, the mini-data set is still not available for further review.

7. PLOS authors have the option to publish the peer review history of their article (what does this mean?). If published, this will include your full peer review and any attached files.

Reviewer #2: No

Reviewer #3: No

---

## [Author Response · Author response to Decision Letter 1]

26 Feb 2020

RESPONSE TO REVIEWER2

1. Comment 2 (analysis section) not been addressed correctly. You can select the optimal cut-off points based on lowest difference between specificity and sensitivity for your cases. (with SPSS or STATA software) The "larger AUC" is not correct for comparing the variables (in the abstract and text). Table 2 should be revised accordingly.

Response to Comment 1: We greatly appreciate the Reviewer’s very insightful and helpful comment. In accordance with the Reviewer’s suggestion, please note that we have now revised the text in the statistical analysis and table 2 regarding the optimal cut-off points with sensitivity and specificity of each indices. In addition, we revised “larger” AU-ROC to “large” AU-ROC in order to avoid misleading. We believe that this new statement adequately addresses the reviewer's comment. (p. 2, lines 15-18, p. 3, lines 3, p 12, lines 5-9, p 15, lines 3-7, p 17, lines 9, p 18, lines 1-2,4-5, p18 lines14-18)

RESPONSE TO REVIEWER3

1. The authors have revised the majority of the manuscript according to the reviewers' suggestions. However, the mini-data set is still not available for further review

Response to Comment 1: We greatly appreciate the Reviewer’s very insightful and helpful comment. We have now added the data set which include the values to use the ROC analysis as supporting information (S1 dataset). We believe that this new information adequately addresses the reviewer's comment.

---

## [Decision Letter · Decision Letter 2]

17 Mar 2020

Anterior and posterior ratio of corneal surface areas: A novel index for detecting early stage keratoconus

PONE-D-19-34557R2

Dear Dr. Kitazawa,

We are pleased to inform you that your manuscript has been judged scientifically suitable for publication and will be formally accepted for publication once it complies with all outstanding technical requirements.

With kind regards,

Yu-Chi Liu, M.D

Academic Editor

PLOS ONE

Additional Editor Comments (optional):

Reviewers' comments:

Reviewer's Responses to Questions

**Comments to the Author**

1. If the authors have adequately addressed your comments raised in a previous round of review and you feel that this manuscript is now acceptable for publication, you may indicate that here to bypass the “Comments to the Author” section, enter your conflict of interest statement in the “Confidential to Editor” section, and submit your "Accept" recommendation.

Reviewer #2: All comments have been addressed

2. Is the manuscript technically sound, and do the data support the conclusions?

Reviewer #2: Yes

3. Has the statistical analysis been performed appropriately and rigorously? 

Reviewer #2: Yes

4. Have the authors made all data underlying the findings in their manuscript fully available?

Reviewer #2: Yes

5. Is the manuscript presented in an intelligible fashion and written in standard English?

Reviewer #2: Yes

6. Review Comments to the Author

Reviewer #2: (No Response)

7. PLOS authors have the option to publish the peer review history of their article (what does this mean?). If published, this will include your full peer review and any attached files.

Reviewer #2: No

---

## [Editor Report · Acceptance letter]

20 Mar 2020

PONE-D-19-34557R2 

Anterior and posterior ratio of corneal surface areas: A novel index for detecting early stage keratoconus 

Dear Dr. Kitazawa:

I am pleased to inform you that your manuscript has been deemed suitable for publication in PLOS ONE. Congratulations! Your manuscript is now with our production department. 

With kind regards,

on behalf of

Dr. Yu-Chi Liu 

Academic Editor

PLOS ONE